# Spermatogonial Stem Cell Transplantation in Large Animals

**DOI:** 10.3390/ani11040918

**Published:** 2021-03-24

**Authors:** Xin Zhao, Weican Wan, Xianyu Zhang, Zhenfang Wu, Huaqiang Yang

**Affiliations:** National Engineering Research Center for Breeding Swine Industry, College of Animal Science, South China Agricultural University, Guangzhou 510642, China; 20182024020@stu.scau.edu.cn (X.Z.); wwc@stu.scau.edu.cn (W.W.); 2010020066@zqu.edu.cn (X.Z.)

**Keywords:** germline ablation, genetic sterility, large animals, livestock, surrogate sire, spermatogonial stem cells, transplantation

## Abstract

**Simple Summary:**

The spermatogonial stem cell (SSC) is the only adult stem cell in males to transmit genetic information to offspring. SSC transplantation (SSCT) is a laboratory technique to regenerate spermatogenesis in recipient males, thus can be used as a novel breeding tool to benefit animal production. Although successful SSCT in rodent models has been established, progress in realizing SSCT in large animals has been limited. Here we discuss what we learned in this area from past experiments and highlight possible directions in developing effective SSCT protocol in large animals.

**Abstract:**

Spermatogonial stem cell transplantation (SSCT) can restore male fertility through transfer of germline between donor and recipient males. From an agricultural perspective, SSCT could be an important next-generation reproductive and breeding tool in livestock production. Current SSCT approaches in large animals remain inefficient and many technical details need further investigation. This paper reviews the current knowledge on SSCT in large animals, addressing (1) donor spermatogonial stem cell (SSC) preparation, (2) recipient male treatment, and (3) SSC injection, homing, and detection. The major studies showing unequivocal evidence of donor SSC-derived spermatogenesis in large animals (mainly in livestock for breeding purpose) are summarized to discuss the current status of the field and future directions.

## 1. Introduction

Spermatogenesis, a highly complex and tissue-coordinated process, maintains sperm production throughout the lifetime of male mammals. The high efficiency of the spermatogenesis system depends on the continuous proliferation and differentiation of spermatogonial stem cells (SSCs) [1,2,3]. SSCs, located on the basement membrane of the seminiferous tubules of the testis of male mammals, have the capacity to both renew themselves to maintain stable populations and differentiate into spermatogenic cells at various developmental stages to form spermatozoa through an orderly, regulated process [4,5,6,7]. SSCs are the foundation of spermatogenesis and male fertility. SSC transplantation (SSCT) is an important application of SSCs for animal reproduction and regenerative medicine. In contrast to the normal process of spermatogenesis, SSCT begins with the migration of donor SSCs from the lumen to the periphery of the seminiferous tubules [8,9]. In this process, the donor SSCs obtained by isolation and purification from testicular tissues are transplanted into the recipient testicular seminiferous tubules, and then colonized into the special micro-environment “niche”. The implanted SSCs can undergo the process of self-renewal and differentiation into spermatogenic cells at all levels, eventually re-establishing donor-derived spermatogenesis [10,11].

For the first time in 1994, Brinster and his colleagues transplanted testicular cell suspensions containing SSCs from fertile donor mice into the seminiferous tubules of the testes of infertile recipient mice. They observed complete donor cell-derived spermatogenesis, which resulted in the formation of mature sperm, and obtained the offspring of donor spermatogonia through natural mating [8,9]. This study opened new avenues for utilization of SSCs to serve the field of biomedicine and agriculture, such as preparation of genetically modified animals, preservation of genetic resources such as endangered species, and treatment of male infertility. In the following decades, SSCT was attempted in various animal species, including pigs [12,13,14,15], cattle [16,17,18], goats [19,20,21], sheep [22,23,24,25], monkeys [26,27,28,29,30,31], dogs [32,33], tree shrews [34], and camels [35]. However, success remains limited and many reports lack unequivocal evidence of true donor-derived spermatogenesis following transplantation. Only a limited number of studies yielded donor SSC-derived embryos and offspring, and those that did, did so at low efficiencies [13,14,19,24,36]. These limitations hinder SSCT application in the agricultural setting. Furthermore, species-specific knowledge is required to establish the optimal protocols for SSC isolation and culture, recipient preparation, and cell transplantation.

The purpose of this review is to provide the latest information in the field of SSCT study in large animal models, focusing on the use of agricultural animals as a breeding tool for surrogacy purposes. We provide an overview of current progress in SSC biology, recipient preparation, and cell transfer methods, and summarize the key attempts and outcomes of SSCT in various large animal models.

## 2. Overview of Spermatogenic Process

SSCs derive from gonocytes, which come from primordial germ cells (PGCs) in early embryos [37,38,39]. During fetal development, PGCs migrate from the base of the allantois into the genital ridges along the hindgut and from here give rise to gonocytes [40,41]. After birth, some of the gonocytes propagate and migrate gradually from the center of the seminiferous cords to the basal lamina of the seminiferous tubules, in which they differentiate into SSCs that drive spermatogenesis [42,43,44,45]. In mice, the transition of gonocytes to undifferentiated spermatogonia occurs between 0 and 6 days postpartum [42]. In pigs, this transition starts at birth and is completed at 2 months of age [43]. However, conversion of gonocytes in cattle occurs at the age of 3–5 months for *Bos taurus* bulls and 9–9.5 months for *Bos indicus* bulls [44,45]. The early onset of spermatogenesis in pigs could partly explain the much more rapid sexual maturity observed in pigs relative to other domestic animals [43].

The stem cell pool in testis is maintained and expanded through self-renewing divisions of SSCs and a series of mitotic divisions that facilitate the transition of SSCs to spermatogonia with different states. The spermatogenic process is generally similar in almost all mammals [46]. In the beginning, spermatogonia is divided into type A, intermediate (In), and type B spermatogonia, which are simply categorized by their amount of heterochromatin. Thereafter, more generations of spermatogonia with different differentiating stages (especially among type A spermatogonia) were discovered to further categorize them [47]. The spermatogonial population consists of undifferentiated and differentiating spermatogonia. The classes of undifferentiated spermatogonial populations is consistent in most mammal species, consisting of type A_single_ spermatogonia (A_s_), A_paired_ spermatogonia (A_pr_) (chains of 2 cells connected by an intercellular bridge), and A_aligned_ spermatogonia (A_al_) (chains of 4, 8, and often 16 cells) generated by successive mitotic divisions. The least differentiated A_s_ spermatogonia are generally considered to be SSCs, while A_pr_ are considered to be progenitor spermatogonia [47]. Broadly speaking, the germ cell populations that can regenerate the spermatogenic lineage following SSCT can all be considered SSCs, which often include gonocytes and partial undifferentiated spermatogonia. Almost all A_al_ cells transition without cell division to A_1_ differentiating spermatogonia [48,49], and then undergo 2–6 generations of differentiating spermatogonia to produce primary spermatocytes. In mice and pigs, the differentiating spermatogonial population is formed by A_1_, A_2_, A_3_, A_4_, In, and B spermatogonia [46]. In cattle and sheep, the differentiating spermatogonial population contains A_1_, A_2_, A_3_, In, B_1_, and B_2_ spermatogonia [46]. Afterward, spermatocytes enter meiosis to generate haploid spermatids, which finally develop to spermatozoa via cytodifferentiation (Figure 1).

Spermatogenesis in mammals occurs in a finely regulated and highly productive manner. The frequency of the spermatogenic cycle is around 10 days and the whole of spermatogenesis is estimated to consume 1–2.5 months (approximately 4.5 cycles) in the mammalian species that were already investigated [50]. Daily sperm production (DSP) per gram of testicular parenchyma is 4–50 × 10^6^/g depending on species [46,50]. In pigs, each spermatogenic cycle lasts 8.6–9.0 days and the entire spermatogenic process takes approximately 40 days, with 24–27 × 10^6^/g DSP [46,51,52]. Humans have a relatively lower efficiency of spermatogenesis with a spermatogenic cycle of 16 days and 4–6 × 10^6^/g DSP [50].

## 3. Preparation of Donor SSCs

Like other adult stem cells, SSCs are very sparse in number. There are only about 35,000 SSCs per testis in adult mice, accounting for about 0.03% of the total number of germ cells in testis [53]. In neonatal pigs, gonocytes only comprise approximately 7% of cells in seminiferous tubules [54]. Pure SSCs are difficult to obtain in primary culture of spermatogonia, as they usually contain a large number of progenitor and other undifferentiated spermatogonia as well as Sertoli cells. The specific molecular marker identifying the bona fide SSCs is debated in the existing literature, and remains undefined in many animal species. Effective SSCT largely depends on the relative abundance of SSCs transplanted into the seminiferous tubules of the recipient’s testis [11,55]. Obviously, increasing the number of SSCs in the donor cells as much as possible will have a positive impact on the colonization and proliferation of SSCs in the recipient’s testis after transplantation. Although there are several strategies to increase the proportion of undifferentiated spermatogonia in the population of germ cells, such as the use of cryptorchid, vitamin-A deficient, or steel mutant mice, these strategies cannot be easily applied in large animals due to the increased complexity of the treatment procedure [56,57,58,59]. In large animals, donor cells usually isolated from neonatal/immature testes contain a higher proportion of gonocytes or undifferentiated spermatogonial populations than those of adults. Although the number of donor cells can be maximized by selecting immature testes, this method cannot ensure a significant increase in the number of SSCs. Therefore, it is necessary to improve the long-term in vitro culture system of SSCs to maintain and expand their number without losing the stem cell potentials. To this end, numerous studies identified several key growth factors that influence the survival, self-renewal, and proliferation of SSCs. For example, Meng et al. revealed the important role of glial cell line-derived neurotrophic factor (GDNF) in determining the fate of undifferentiated spermatogonia in mouse models. In mice, knocking out a single allele of GDNF resulted in ablation of the stem cell reservoir and blocked spermatogenesis, leading to severe fertility defects. In contrast, mice overexpressing GDNF showed accumulation of undifferentiated spermatogonia, resulting in the formation of non-metastatic testicular tumors [60]. Soluble GDNF-family receptor α-1 (GFRα1) can act in synergy with GDNF to enhance in vitro expansion of SSCs in mice and large mammals, which suggests a key role for GDNF signaling in SSC self-renewal [61,62,63]. In addition, other factors, such as basic fibroblast growth factor (bFGF), colony stimulating factor 1 (CSF1), leukemia inhibitory factor (LIF), epidermal growth factor (EGF), and insulin-like growth factor 1 (IGF1), also play important roles in SSC maintenance and growth, and are used synergistically as medium supplements for in vitro culture of SSCs [62,63,64,65,66,67,68]. Currently, long-term in vitro culture systems for SSCs have been established for mice [67,68,69,70], rats [71,72], and hamsters [73].

While researchers had success with these long-term in vitro cultures in rodent models, several teams only achieved short-term culture of large animal SSCs under similar culture conditions. Goel et al. used a lectin, *Dolichos biflorus* agglutinin, to specifically enrich pig primitive germ cells, and cultured purified gonocytes in medium containing 10% FBS for a short time [74,75]. However, the number of gonocytes gradually decreased with every passage, which could be attributed to the numerous undefined factors that induce cell differentiation present in FBS. In addition, overwhelming growth of contaminating somatic cells can monopolize the nutrients needed for SSCs to sustain proliferation. Subsequently, the researchers tried to culture SSCs in vitro, with low-concentration serum or without serum, either in a feeder-free condition or using mitomycin C-treated Sertoli cells as feeder. Zheng et al. found in a separate study that in vitro proliferation of porcine SSCs could be maintained for 1 month under the condition of 1% FBS and supplementation of growth factors (bFGF and EGF) [76]. Thereafter, they found that the undifferentiated porcine spermatogonia could proliferate in vitro for at least 2 months on neonatal Sertoli cells as feeder, using DMEM/F12 culture medium supplemented with 10% serum-free supplement KSR and 4 growth factors (GDNF, GFRα1, bFGF, and IGF1) [63]. Recently, Oatley’s team reported a long-term in vitro culture system for bovine SSCs, in which they used bovine fibroblasts as a feeder to successfully maintain undifferentiated bovine spermatogonia for 2 months. The morphological characteristics of the germ cell clumps resembled those in mice, rats, rabbits, and hamsters [77]. So far, the in vitro culture system of large animal SSCs was reported in pigs [63,74,75,76], cattle [77], and goats [78], but whether these primary cultures contain true SSCs is unclear. There remains a lack of unequivocal evidence that spermatogenesis can arise from these cultured SSCs when evaluated by the transplantation approach. Therefore, further investigation is required to improve the long-term in vitro cultivation system of large animal SSCs and establish specific strategies to identify the true SSCs supporting regenerated spermatogenesis after SSCT.

## 4. Preparation of Recipients

Another key issue that restricts the effective application of SSCT in large animals is the preparation of ideal recipients that can support re-established donor-derived spermatogenesis following cell transfer. The ideal recipients used in SSCT should have endogenous spermatogenesis ablation or suppression while retaining complete somatic cell structure and function [79]. Endogenous SSC depletion could spare stem cell niches to accommodate the transplanted donor SSCs [79,80]. SSCT application in animal production also requires a maximal exclusion of endogenous SSCs to avoid production of offspring of endogenous origin. To address this issue, a variety of strategies for germline ablation to generate sterile recipients are available, such as chemotherapy [8,9,10], irradiation [81,82], heat shock treatment [83,84], or the use of genetically defective animals with congenital deletion of endogenous spermatogenesis [8,9,36].

The chemotherapeutic agent busulfan can effectively ablate endogenous germline cells to support SSCT in mouse models. However, its effectiveness in preparing large animal recipients is limited. Busulfan usually cannot ablate the germ cell pool completely, and different degrees of recovery of endogenous germ cells are almost inevitable. This could preclude the homing of transplanted SSCs. Busulfan is also systemically toxic, causing bone marrow suppression and anemia [85]. A species-specific sublethal dose of busulfan is required to ablate endogenous spermatogenesis. It is thus difficult to determine a standard dose for optimal germline ablation in different animal species and individuals. Likewise, testicular irradiation is also an ineffective method for germline ablation in large animals. Irradiation effects vary according to species, animal age, and irradiation dose [16,24]. Irradiation also severely damages the viability of Sertoli cells, destroys the structure of seminiferous tubules, and causes permanent sterility [86,87]. In addition, recovery of endogenous SSCs is common in follow-up analysis [88]. Another factor that limits the application of irradiation to large animals is the need for specialized facilities and equipment. Unequivocal evidence of successful SSCT in busulfan or irradiation-treated large animal recipients was reported in limited studies with a low efficiency [13,14,24]. Among them, generation of donor-derived offspring was only reported in one study in sheep, in which the re-established spermatogenesis in irradiated rams produced less than 15% of sperm and subsequent offspring with donor haplotype by artificial insemination (AI) [24].

The latest gene editing technology provides new options for eliminating endogenous SSCs in animals. Using gene editing technology to knock out (or inactivate) a certain gene, which only expresses in male germ cells or plays a key role in their survival and development can selectively eliminate endogenous SSCs without affecting the integrity of somatic support cell structure and function, thus providing sufficient niches for colonization and differentiation of transplanted SSCs [89,90]. The most significant advantage of genetic sterility over traditional drug or irradiation ablation is that the evacuation of the endogenous germline is complete and stable, without recovery over time. The sterile phenotype is also inheritable and can therefore be obtained by mating heterozygous animals without having to treat each recipient individually. Early studies in mice provided a proof-of-concept that spontaneous mutations at the W locus of the c-Kit gene block the migration of PGCs to the gonads during embryonic development and lead to the depletion of germ cells, while leaving the complete structure and function of seminiferous tubules intact, making this mouse model a suitable recipient for SSCT [48]. Transplanted germ cells can effectively colonize and initiate spermatogenesis in the testes of W mutant mice, as well as restore natural fertility [8,9,10]. NANOS2 is a male PGC-specific gene essential for fetal male germ cell development. Knockout of NANOS2 in mice results in male sterility due to the complete loss of spermatogonia [91]. Translation of this gene modification strategy to livestock (pigs, goats, and cattle) [36,89,90] can phenocopy knockout mice with male specific germline ablation while retaining intact Sertoli cells. More importantly, both heterozygous knockout males and homozygous knockout females of NANOS2 alleles are fertile, which allows us to obtain knockout males easily through cross-breeding, and to sustain an SSCT-adapted sterile male population with an inheritable phenotype [36].

## 5. SSC Injection, Homing, and Detection

Three routes are usually used to inject donor cells into recipient testes. SSCs can either be injected into the efferent ducts connecting the rete tubules and epididymis, the rete tubules formed by the convergent ends of all seminiferous tubes, or the seminiferous tubes themselves [10,92]. In most mammals, testicular lobules are closed compartments formed by the tunica albuginea extending inward and separating the testes, where the seminiferous tubules exist in a highly coiled and tightly packed manner [92,93,94]. A precise injection of donor cells into the lumen of seminiferous tubules may require multiple incisions in the tunica albuginea, which increases the complexity of the manipulation and injury to the testis of recipient [10]. Also, the potential increase of the intratubular pressure during the injection process may cause rupture of the membrane [92]. With regard to efferent tubules, the efferent tubules in rodents usually merge and form a single common efferent tubule connected to the epididymal head, providing easy access for injection into the common route. However, in large animals, multiple efferent tubules enter the epididymal head in parallel, which may complicate injection through the efferent tubule [92]. The rete testis has a distinct structure in testis, making it easily distinguishable from the surrounding tissues and offering more practical access to the seminiferous tubules [12]. However, the anatomical structure of rete testis differs among species. In rodents, including mice, rats, and hamsters, the rete testis is located closer to the subcapsular area, while in most other species, such as cats, dogs, rams, bulls, boars, and monkeys, the rete testis is located longitudinally and deeply in the center of the testis [10,92,95]. Therefore, the rete testis is more difficult to access and visualize in large animals compared with rodents, and imaging techniques such as ultrasound imaging are required for a precise injection. Taken together, introduction of donor germ cell suspension into the rete testis under ultrasound guidance or surgical dissection is nonetheless a feasible pathway for most large animals. SSCT into rete testis allows an infusion of a large volume of cell suspension, which can be injected at multiple sites to ensure a higher efficiency of cell transfer.

The homing ability of stem cells allows them to migrate and colonize existing niches. SSC homing refers to a process in which the transplanted SSCs pass through the blood-testis barrier (BTB) formed between two supporting cells in the opposite direction toward open niches on the basement membrane [96]. Homing is an inefficient process due to the inability of some cells to pass through the BTB. SSCs require approximately 1 week to colonize the stem cell niches upon transplantation and the homing efficiency is approximately 12 % in mice [97]. In this regard, SSCT in immature testis prior to the formation of a stable BTB, which occurs between 10 and 16 days in mice [98] and before 120 days in pigs [99] after birth, could improve the efficacy of SSC homing. The efficiency of SSC colonization is 4–10 times higher in immature testis than in mature testis and also significantly decreases as donor males age [2,100,101,102]. Therefore, SSCs from young donors could maximally colonize the young SSC niches. As the only somatic cells within the seminiferous tubules, Sertoli cells are essential in establishing SSC niches, and consequently markedly influence the homing efficiency of transplanted SSCs. The volume density of Sertoli cells is relatively lower in small rodents than in large animals (15% in mice vs. 25% in pigs, according to previous reports) [46,103]. A lower proportion of Sertoli cells in seminiferous epithelium generally corresponds to a higher Sertoli cell efficiency, supporting a higher rate of colonization and spermatogenesis after SSCT [46,103]. This could be one factor accounting for the low efficiency of SSCT in large animals.

Application of SSCT in livestock as a breeding tool requires allogeneic or homologous transplantation. This requires consideration of the genetic and immune compatibility between recipients and cell donors. On the one hand, the success of SSCT seems to depend on the availability of recipients that are either genetically compatible with the donor animal, or have innate immune deficiency or receive immunosuppressive treatment [16,104]. On the other hand, the phenomenon of immune rejection of incompatible donor cells has not been widely observed in recipients with intact immune system functions, such as pigs [13,14,15], cattle [18], goats [19], and sheep [24]. The immune tolerance of recipient testis to donor SSCs could result from immune-privilege provided by the environment of the testis. Multiple mechanisms could promote this immune privilege, including the presence of the BTB, which prevents immune components from contacting implanted cells through tight junctions between adjacent Sertoli cells, and the variety of immunosuppressive factors secreted by somatic cells to suppress immune rejection [96,105,106]. Recent advances by the use of genetically modified sterile surrogates showed that allogeneic SSCT without immune suppression is tolerated in mice, pigs, and goats, and allogeneic SSC-mediated spermatogenesis is robust enough to produce persistent natural fertility in mouse models [36]. However, we should note that even though allogeneic SSCT can restore a natural fertility in mice, the recipient males typically produce significantly decreased offspring per litter [36]. The impaired fertility following allogeneic SSCT implies the possible side effect of immune rejection between recipients and donor SSCs, which requires further investigation to elucidate the activity of immune components in SSCT.

Tracing the colonized SSCs and re-established spermatogenesis that results from SSCT is generally challenging; most previous attempts in large animals involved the administration of cytotoxic treatments to recipients, resulting in residual endogenous spermatogenesis. Some studies used fluorescent dyes (such as PKH26, PKH67, and carboxyfluorescein diacetate succinimidyl ester (CFDA-SE)) to monitor the location of the donor cells and the duration of their persistence in the seminiferous tubules of recipient testes [12,17,20,21,22,24]. This approach is imprecise, as a mixed suspension of testicular cells or roughly purified germ cells are labeled in these reports. Presence of fluorescence-positive cells in seminiferous tubules in recipients does not therefore necessarily reflect the engraftment of donor SSCs when this technique is used. Moreover, the gradual dilution of fluorescent dyes as cells divide, in addition to their biological instability, rarely provides evidence for the long-term persistence and development of donor cells [107,108]. Another method of tracing the presence and development of donor SSCs is to use transgene (*Escherichia coli* β-galactosidase gene (lacZ) or enhanced green fluorescent protein (EGFP)) to label donor cells. Generally, researchers use lentiviral or adeno-associated viral (AAV) vectors to deliver reporter genes to donor SSCs for transplantation. Using this method, recipient ejaculate harboring donor-derived transgenic sperm can be detected after SSCT in pigs [13,14], goats [19,109,110], tree shrews [34], and monkeys [29]. Furthermore, in vitro fertilization (IVF) of the donor-derived transgenic sperm or natural mating of transplanted recipients successfully generated embryos in pigs [13,14], goats [110], and monkeys [29], as well as live transgenic progeny in goats [19] and tree shrews [34], validating the re-establishment of complete donor-derived spermatogenesis. The most critical problem associated with viral transduction is that the intact virus particles that remain following in vitro transduction can be co-transferred into recipient testis and infect endogenous germ cells or somatic cells, thus interfering with the accurate identification of transgene-positive semen samples of true donor origin. Furthermore, sperm are not the only cells that can be present in an ejaculate, thus simply detecting transgenic DNA does not unequivocally show that it originated in sperm derived from transplanted SSCs [111,112]. Likewise, the microsatellite marker analysis that distinguishes donor-derived and endogenous spermatogenesis also cannot exclude the possibility of interference of transplanted somatic cells or undifferentiated germ cells present in the semen [18,32,33]. Taken together, the results of these tracking approaches require discreet analysis, and combinational use of multiple analyses is necessary to confirm true spermatogenesis from transplanted SSCs.

Overall, the current experimental approaches to donor SSC-derived spermatogenesis remain inefficient; low proportions of donor genotypes are readily identifiable in recipients in the existing literature, especially in large animals. We here draw a brief schematic diagram outlining the typical procedure of SSCT in laboratory mouse models and large animals (Figure 2). The procedure is similar between different animals, but some technical details are more complex and remain unaddressed in large animals.

## 6. Unequivocal Evidence of Successful SSCT in Large Animals

In this section, we summarize the clear evidence from previous experiments that donor-derived spermatogenesis occurs following SSCT in large animals. The studies that only included single lines of evidence, such as repopulation of transplanted donor cells identified by fluorescent dye labeling or histological assessment, or genotyping of ejaculate by microsatellite markers or transgenic DNA, are not included here. In addition, SSCT in monkey models is also excluded, as these models are usually developed as a strategy to address human infertility caused by chemotherapy or radiotherapy. In these approaches, testicular tissue cryopreservation and autologous transplantation are required, which differs from SSCT application in livestock for breeding purposes. The technical details of these reports are listed in Table 1 for an easy overview.

In the first example referred to in Table 1, Zeng et al. transplanted purified pig spermatogonia transfected with lentivirus or AAV into the testis of normal recipient boars with intact germline or recipient boars treated with busulfan to eliminate the endogenous germ cells at 12 weeks of age, through ultrasound-guided cannulation of the rete testis. The authors detected transgene starting from 9–11 months after transplantation, and some 5-year-old recipient boars were still positive for transgene in ejaculates. Notably, the donor-regenerated transgenic sperm functioned normally to fertilize eggs and form transgenic embryos [13]. Subsequently, Kim et al. transplanted lentivirus (harboring EGFP)-transfected SSCs into the testes of 12- to 16-week-old recipients born from sows treated with busulfan. This study obtained viable EGFP-positive spermatozoa from 2 out of 6 recipients and further obtained EGFP-positive embryos originated from transplanted SSCs by ICSI [14].

Honaramooz et al. also demonstrated success in goats, transplanting the mixed testicular cells isolated from transgenic goat testes expressing human alpha-1 antitrypsin into untreated immature wild-type recipient goats, using the rete testis injection method. The presence of transgene was detected in the ejaculate of 2 recipients, and transgenic offspring were obtained through natural mating [19]. This study demonstrated that using non-enriched germline stem cells for allogeneic SSCT in non-ablated immunocompetent recipient goats is feasible to support production of donor-derived live animals by natural fertility. The same team also transduced goat testicular cells with AAV carrying the EGFP transgene and transplanted them into the testis of recipient goats with testicular irradiation. They collected the semen of recipients starting from 5 months after transplantation and found that 37 and 35% of ejaculates were positive for EGFP, as detected by PCR, over an 18-month period. They further obtained 15/155 and 12/121 EGFP-positive embryos by IVF using the semen of 2 recipients, validating a functional sperm differentiated from the donor cells [110].

Similar strategies were also employed in sheep. Herrid et al. transplanted isolated sheep testicular cell suspension into recipient testes (from the same or different breeds) treated with 9 or 15 Gy irradiation. The semen collected starting from 6–10 months after transplantation showed donor-derived DNA in some ram recipients. Production of donor-derived sperm was detected beginning at 15 weeks after SSCT. Animals receiving 15 Gy irradiation generated more significant donor-derived sperm than 9 Gy and non-irradiated groups. The proportion of donor sperm in 15 Gy irradiation recipients was 9.7% on average, with the highest percentage (30%) identified at 40 weeks. Subsequently, the semen of transplanted recipients was used to perform AI on 151 ewes whose estrus cycles were synchronized. Microsatellite marker analysis of the newborn lambs showed that 4 donor-derived lambs (7.6% of progeny) resulted from a recipient in Merino-to-Merino transplantation, and 6 lambs (14.6% of progeny) were sired by donor-derived Border Leicester sperm produced in a Merino recipient ram [24]. These collective results support the premise that preparing recipient animals with appropriate doses of radiation and allogeneic SSCT between breeds is feasible in sheep, showing promise for the use of SSCT as a replacement for AI for extensive sheep production.

Germline ablation through genetic modification substantially advances SSCT progress in large animals. Genetic sterility can completely and inheritably remove endogenous germ cells without impairing somatic supporting cells and testicular function. The sterile condition in animals is stable without recovery of endogenous spermatogenesis over time. Demonstrating this, Park et al. generated NANOS2 knockout pigs by directly injecting the CRISPR reagent into the cytoplasm of pig embryos to produce inactivating mutations in the NANOS2 coding region. The resultant NANOS2 knockout pigs mimicked the phenotype of knockout mice, showing a loss of SSCs while retaining intact seminiferous tubules. Serum testosterone secretion level was not different from that of normal pigs [89]. The researchers then carried out SSCT with NANOS2 knockout boars as recipients. Transplantation was performed twice with donor testicular single cell suspension, and enriched undifferentiated spermatogonia (about 70% based on immunostaining for the spermatogonia marker ZBTB16) were introduced into the rete testis of the recipient testis at the age of 4 months and 14–15 months, respectively. After transplantation, semen was regularly collected from all NANOS2 knockout recipients starting from about 6 months of age. Complete NANOS2 allele was detected in the DNA of boar ejaculates, and sperm was detected in epididymal flushing for as long as 2 years after transplantation, confirming stable, long-term re-established donor-derived spermatogenesis. In addition, histological examination showed that about 15% of seminiferous tubules restarted spermatogenesis, and sperm was observed in epididymal tubules. This study also demonstrated that donor-derived spermatogenesis did not reach the level equivalent to that of normal wild-type males, which is key to obtaining fertility in a natural breeding environment [36]. Using a similar genetic modification strategy, Fan et al. established NANOS2 knockout goats by somatic cell nuclear transfer (SCNT). The knockout bucks developed normally from birth to adulthood. Histological examination revealed the absence of germline cells in testicular cross-sections, while the seminiferous tubules remained intact [90]. When NANOS2 knockout bucks were 4 months old, enriched undifferentiated spermatogonia (about 60% for ZBTB16 immunostaining) isolated from the donor male testes were transplanted into the rete testis under ultrasound guidance. The author reported that round sperm cells appeared in the ejaculation of 3 recipients about 85 days after transplantation, and motile sperm appeared for the first time in one recipient at 136 days after transplantation. These sperm had complete NANOS2 alleles, indicative of donor cell origin [36].

## 7. Summary

Modern livestock production pursues high-efficiency propagation of elite animals with desirable genetics. To this end, alternative and novel reproductive techniques applicable to livestock have been subjects of ongoing study for multiple generations, from natural breeding to AI, SCNT, and then SSC manipulation. The SSCT-based “surrogate sires” concept proposed by some active investigators in this field can rapidly and selectively proliferate and disseminate gametes from desirable sires (without having to use the sires themselves), significantly accelerating the progress of livestock breeding and production [113,114,115]. The key elements to achieving this potential involve lengthy in vitro culture and large-scale proliferation of SSCs, production of germline-ablated recipient males, and effective cell transfer and homing. Currently, SSCT in large animals is feasible, but limited success has been achieved. A stable and efficient, but also species-specific, SSCT protocol in large animals has yet to be established.

In light of the recent success of NANOS2-knockout large animal models in SSCT application, cultivating such genetically sterile recipient males could be a simple and uniform strategy to facilitate effective SSCT in large animals. Many other genes associated with germline maintenance and development can be explored and modified to create ideal recipient male models. In addition, an efficient genetic modification technique in SSCs should be investigated to support generation of SSC-derived transgenic or gene-edited large animals, overcoming the low efficiency, and high cost of embryo-based genome modification approaches including pronuclear injection and SCNT.

## Figures and Tables

**Figure 1 animals-11-00918-f001:**
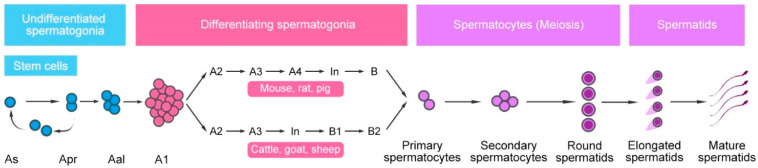
Mammalian spermatogenesis model. Undifferentiated spermatogonia contain type A_s_, A_pr_, and A_al_ spermatogonia. Among them, self-renewing proliferation of early spermatogonia, mainly including A_s_ and probably partial A_pr_, sustains a stem cell pool, which gives rise to spermatogonia progenitors and initiates spermatogenesis. A_al_–A_1_ transition represents the start of spermatogonial differentiation. In this process, type A_1_ spermatogonia undergo a fixed number of mitotic divisions to produce primary spermatocytes. The numbers of differentiated spermatogonial generations are species dependent. Afterward, spermatocytes enter meiotic replication to produce round spermatids, which finally develop to mature spermatids via cytodifferentiation.

**Figure 2 animals-11-00918-f002:**
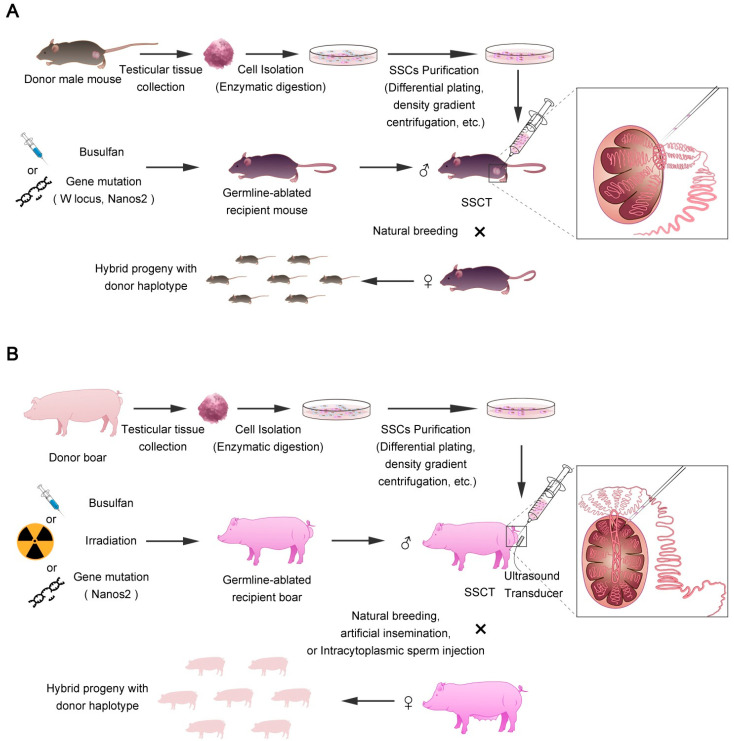
Spermatogonial stem cell transplantation (SSCT) procedure in mouse models and livestock species. Schematic showing the typical steps involved to produce donor SSC-derived progeny through SSCT in mice (**A**) and pigs (**B**). Testicular cells are isolated from the testis of donor male animals by collagenase digestion. Afterward, spermatogonia are enriched through multiple cycles of differential plating. The purity of spermatogonia can be further enhanced through discontinuous Percoll density gradient centrifugation. The enriched spermatogonia are adjusted to suitable concentration for transplantation. To ensure an effective SSCT, ablation of endogenous Spermatogonial stem cells (SSCs) of recipient males is required. In this regard, busulfan treatment is generally effective for mouse SSC ablation, but remains inefficient in large animals. Other ablative methods, such as irradiation of testis, also has limited effect on endogenous SSC removal to support an effective SSCT. Recently developed genetic modification technology can create genetically sterile animals with a complete SSC ablation but preserving intact testicular supporting cell structure, therefore offering an alternative approach to generate SSC-ablated recipient males. In mice, transplanted cells are usually injected through efferent tubules, whereas rete testis is the easy route for injection of spermatogonia in some large animals. Successful SSCT can re-establish spermatogenesis of recipient animals to generate donor-derived sperm, which can fertilize eggs to produce offspring harboring donor gene alleles.

**Table 1 animals-11-00918-t001:** Summary of SSCT in large animals.

Species	Donor SSCs	Recipient Treatment	Transplantation Type	Injection Sites	Injection Volume and Total Cell Number per Testis	Detection Methods of Transplanted Cells	Transplantation Results	Spermatozoa Condition	Embryo Condition	No. of Offspring with Donor Haplotype	References
Pig	10–12-week-old piglets; purification through Staput velocity sedimentation; AAV and lentiviral transduction	12-week-old recipients born to busulfan-treated sows and age-matched normal boars without treatment	Allogeneic	Rete testis	3–5 mL; 0.2–1.1 × 10^9^ testicular cells or 0.2–0.4 × 10^8^ enriched spermatogonia	Generation of donor-derived embryos with EGFP expression	Production of transgenic embryos through IVF	0–54.8% and 0–25% EGFP-postive ejaculates from recipients transplanted with germ cells transduced with AAV and lentivirus, respectively	Generated by IVF; 1/11 and 18/28 transgenic embryos from 2 recipients receiving AAV-transduced germ cells; 1/13 and 5/19 transgenic embryos from 2 recipients receiving lentiviral-transduced germ cells	NA	[13]
Pig	12–16-week-old male pigs; purification through discontinuous Percoll density gradient and laminin-coated dishes; lentiviral transduction	12–16-week-old recipients born to busulfan-treated sows	Allogeneic	Rete testis	NR	Generation of donor-derived embryos with EGFP expression	Production of EGFP-expressing embryo by ICSI	2/6 recipients produce transgenic ejaculates; sperm is morphologically normal but concentration is 1/8 of age-matched normal pigs; at least 1 in 100 sperm is estimated to carry the transgene	Generated by ICSI; presence of EGFP fluorescence in embryos	NA	[14]
Pig	Unpurified testicular cells or enriched spermatogonia through differential plating from prepubertal donor boars	NANOS2 knockout male pigs at ∼4 and 14–15 months of age for twice transplantation	Allogeneic	Rete testis	NR; 1–2 × 10^6^ cells/mL	Generation of normal sperm; Genotyping for NANOS2 gene	Repopulation of seminiferous tubules and generation of donor-derived motile sperm up to ~2 years after transplant	Morphologically normal and motile sperm persists in the ejaculate for >200 d, while nontransplanted NANOS2 knockout males remain azoospermic	NA	NA	[36]
Goat	3- and 3.5-years-old transgenic goats carrying the human α1-antitrypsin gene; no enrichment for spermatogonia	~4-month-old wild-type goats without any treatment	Allogeneic	Rete testis	~5 mL; 190–640 × 10^6^ cells	Generation of donor-derived transgenic goats	Generation of sperm carrying the donor-derived transgene; mating of recipient resulted in donor-derived transgenic offspring	2/5 recipients habor transgenic sperm; at least 1 in 50 sperm is estimated to carry the transgene	NA	1/15 offspring is positive for transgene	[19]
Goat	8–11-week-old dairy goats; no enrichment for spermatogonia; AAV-EGFP transduction	4-month-old male dairy goat kids subjected to fractionated testicular irradiation of 3 × 2 Gy at 4 weeks of age	Allogeneic	Rete testis	NR; 100–500 × 10^6^ cells	Genotyping of transgenic EGFP	Presence of EGFP in recipent sperm and IVF embryos generated with the semen from recipient goats	37 and 35% of EGFP-positive ejaculates from 2 recipient goats over an 18-month period	Generated by IVF; 15/155 and 12/121 embryos from semen of 2 recipients are EGFP positive	NA	[110]
Goat	Prepubertal donor bucks; purification through differential plating	4-month-old NANOS2 knockout cloned bucks	Allogeneic	Rete testis	NR; 1–2 × 10^6^ cells/mL	Generation of normal sperm; Genotyping for NANOS2 gene	Motile sperm with normal morphology and an intact NANOS2 allele	Presence of morphologically normal and motile sperm with donor origin	NA	NA	[36]
Sheep	Merino, Border Leicester, or crossbred rams with scrotal circumferences ranging from 13 to 25 cm; no purification of spermatogonia	Merino rams with scrotal circumferences ranging from 21 to 25 cm; Germline ablation by irradiation with doses of either 9 or 15 Gy	Allogeneic	Rete testis	5 mL; 180–230 × 10^6^ cells	generation of donor-derived sperm and offspring confirmed by microsatellite marker assay	Presence of donor sperm in recipient ejaculates; generation of donor-derived lambs by AI using sperm from SSCT	5/5 recipients in 15-Gy irradiation group are positive for donor DNA in ejaculates, with averagely 9.7% of donor sperm in recipient ejaculates	NA	4/52 (7.6%) and 6/41 (14.6%) lambs with donor haplotype from recipient semen following SSCT between same breed and different breeds, respectively	[24]

AAV, adeno-associated virus; IVF, In vitro fertilization; ICSI, Intracytoplasmic sperm Injection; EGFP, Enhanced green fluorescence protein; NA, not applicable; NR, not reported.

## Data Availability

Data sharing not applicable.

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
