# Peer review of "Spermatogonial Stem Cell Transplantation in Large Animals"

_animals, 2021, doi:10.3390/ani11040918_

Round 1

Reviewer 1 Report

The revised version seems to be improved, however in the current version some of the references in Tab1 got messed up and now do not make sense anymore.

Also the question whether the heterologous SSCT can fully restore fertility is not convincinly clarified. Ciccarelli et al. PNAS 2020 showed that mice, which underwent a heterologous SSCT  (supplement) typically do not product more than 3 offspring per litter, whereas normal littersizes in this breed ~ 14! Suggesting that colonization and formation of functional spermatozoa worked, but seem to be limited. The statement that no immune-reactions have been found in mice and livestock after SSCT seems to reflect more the lack of investigation, than a thourough investigation of chronic and long-term consequences.

Author Response

The revised version seems to be improved, however in the current version some of the references in Tab1 got messed up and now do not make sense anymore.

- I have carefully checked the content of Table 1 and rearranged the references if they are not corrected cited.

Also the question whether the heterologous SSCT can fully restore fertility is not convincinly clarified. Ciccarelli et al. PNAS 2020 showed that mice, which underwent a heterologous SSCT (supplement) typically do not product more than 3 offspring per litter, whereas normal littersizes in this breed ~ 14! Suggesting that colonization and formation of functional spermatozoa worked, but seem to be limited. The statement that no immune-reactions have been found in mice and livestock after SSCT seems to reflect more the lack of investigation, than a thourough investigation of chronic and long-term consequences.

- Yes, in whatever aspects immune components would have more or less activity to transplantation, even though testis is considered as an organ of immune privilege. Many publications also reported the side effect of immune rejection to SSCT. We thus changed our discussion to tender previous statement as follows,

However, we should note that even though allogeneic SSCT can restore a natural fertility in mice, the recipient males typically produce significantly decreased offspring per litter [36]. The impaired fertility following allogeneic SSCT implies the possible side effect of immune rejection between recipients and donor SSCs, which requires further investigation to elucidate the activity of immune components in SSCT”.

Reviewer 2 Report

Dear author,

This review summarizes the information about spermatogonial stem cell transplantation in large animals, and focuses of donor SSCs and recipient male preparation, and SSC injection, homing and detection. The manuscript is well written and structured, and collects the main and most current information about this topic. Only minor revisions are necessary:

  • Lines 85, 87, and 89: the authors should define A1, A2, A3, A4, B1, and B2.
  • Line 146: the authors should define GDNF and GFRα
  • Line 183: the authors should define AI acronym.
  • Lines 193, 197 and others: The authors should unify the name of the genes. Genes are usually named in lowercase and italics.
  • Line 243: “vs” must be written in italics.
  • Lines 258-259: Add reference that supports the sentence “More importantly,…mice, pigs, and goats”.
  • Lines 268-272: Add reference that supports this sentence.

Author Response

This review summarizes the information about spermatogonial stem cell transplantation in large animals, and focuses of donor SSCs and recipient male preparation, and SSC injection, homing and detection. The manuscript is well written and structured, and collects the main and most current information about this topic. Only minor revisions are necessary:

Lines 85, 87, and 89: the authors should define A1, A2, A3, A4, B1, and B2.

- We added a sentence to define them in this paragraph as follows,

In the beginning, spermatogonia is divided into type A, intermediate (In), and type B spermatogonia, which are simply categorized by their amount of heterochromatin. Thereafter, more generations of spermatogonia with different differentiating stages (especially among type A spermatogonia) were discovered to further categorize them.

Line 146: the authors should define GDNF and GFRα

- The full names of these genes has been defined before, such as glial cell line-derived neurotrophic factor (GDNF) (Line 125), and GDNF-family receptor α-1 (GFRα1) (Line 130).

Line 183: the authors should define AI acronym.

- Full name has been added (artificial insemination).

Lines 193, 197 and others: The authors should unify the name of the genes. Genes are usually named in lowercase and italics.

-Thank you for your comment. We have checked and corrected some gene names in the revised manuscript. We have used the gene nomenclature according to NCBI gene bank. As we usually used genes from large animals such as pigs. Their gene names usually use capitalized letters, different from mouse gene names in format. For example, we use NANSO2 for pig gene, and Nanos2 for mouse.

Line 243: “vs” must be written in italics.

- It has been corrected.

Lines 258-259: Add reference that supports the sentence “More importantly,…mice, pigs, and goats”.

- Thank you for your comment. This part has been rewritten to add more content to discuss the immune condition in SSCT. The related references have been included if necessary.

Lines 268-272: Add reference that supports this sentence.

- Some previous publications have discussed this point. We have included references to support this statement as follows,

Parish, C. R. Fluorescent dyes for lymphocyte migration and proliferation studies. Immunology and cell biology 1999, 77(6), 499-508, doi: 10.1046/j.1440-1711.1999.00877.x.

Herrid, M.; McFarlane, J.R. Application of testis germ cell transplantation in breeding systems of food producing species: a review. Animal biotechnology 2013, 24(4), 293-306; doi: 10.1080/10495398.2013.785431.

Giassetti, M.I.; Ciccarelli, M.; Oatley, J.M. Spermatogonial stem cell transplantation: insights and outlook for domestic animals. Annu Rev Anim Biosci 2019, 7, 385-401, doi: 10.1146/annurev-animal-020518-115239.

Reviewer 3 Report

Zhao et.al broadly summarized the development of spermatogonial stem cell transplantation technology in rodents and domestic animals. However, I didn’t see any novel points inspired by the authors since nearly all of the works mentioned in this review have already been summarized by previous works published recently (Reproduction. 2014 Feb 3;147(3):R65-74., Annu Rev Anim Biosci. 2019 Feb 15;7:385-401., Hum Reprod Update. 2020 Apr 15;26(3):368-391.) which made this review unnecessity. Some comments are listed below.

Although it has demonstrated in other works, a well-designed figure of spermatogenesis for the animals mentioned in the second section could be helpful to the readers to understand the morphological and molecular difference among species.

In Line 136-137: “could be attributed to the numerous undefined factors that induce cell differentiation present in FBS.” The authors could specify some kind of factors that induce the differentiation process.

In line 151, the authors stated that “…whether these primary cultures contain true SSCs is unclear”, which indicated that checking molecular markers of in vitro “SSCs” is not enough without the spermatogenesis evaluation. However, then they stated that “find species-specific molecular markers” is required for further investigation which contradicted their previous statement.

Author Response

Zhao et.al broadly summarized the development of spermatogonial stem cell transplantation technology in rodents and domestic animals. However, I didn’t see any novel points inspired by the authors since nearly all of the works mentioned in this review have already been summarized by previous works published recently (Reproduction. 2014 Feb 3;147(3):R65-74., Annu Rev Anim Biosci. 2019 Feb 15;7:385-401., Hum Reprod Update. 2020 Apr 15;26(3):368-391.) which made this review unnecessity. Some comments are listed below.

- Thank you for your valuable comments. I admit your criticism is reasonable. However, I hope the editors and reviewers kindly allow us to review this field, as our team definitely works in this field and recently begins to give some outcomes for this (Zhang M, Biol Open. 2021; Zhang X, Livestock Sciences, 2021). Our team has different background (genetic modification of livestock) thus may have some novel contributions to this area later.

Although it has demonstrated in other works, a well-designed figure of spermatogenesis for the animals mentioned in the second section could be helpful to the readers to understand the morphological and molecular difference among species.

- Thank you for your valuable suggestion. We have added a new figure (Figure 1) to demonstrate the general process of spermatogenesis of rodent and livestock.

In Line 136-137: “could be attributed to the numerous undefined factors that induce cell differentiation present in FBS.” The authors could specify some kind of factors that induce the differentiation process.

- It is well known that FBS contains differentiating factors for embryonic and adult stem cells. We have grown SSCs in FBS and the cells become morphologically and phenotypic fibroblasts. This is why many culture systems for stem cells used KSR (Knockout serum replacement with defined factors), not FBS. These differentiating factors have many types in serum, such as BMP or dexamethasone (based on embryonic stem cell studies). It is hard for us to precisely specify differentiating factors in FBS for SSC, as few reports were found in this area. We thus did not add related content for this. However, the statement of “the numerous undefined factors that induce cell differentiation present in FBS” is correct.

In line 151, the authors stated that “…whether these primary cultures contain true SSCs is unclear”, which indicated that checking molecular markers of in vitro “SSCs” is not enough without the spermatogenesis evaluation. However, then they stated that “find species-specific molecular markers” is required for further investigation which contradicted their previous statement.

- Thank you for point this discrepancy in description. We have revised this part as follows,

Therefore, further investigation is required to improve the long-term in vitro cultivation system of large animal SSCs and establish specific strategies to identify the true SSCs supporting regenerated spermatogenesis after SSCT.

Reviewer 4 Report

The manuscript is well written and summarizes the current state of this topic detailed and nicely.

Figure 1: Here, a comment / hint could be included that purified donor SSCs are often propagated and (genetically) manipulated in an in vitro culture system before they are transplanted into a recipient testis.

Not all abreviations used in the manuscript are explained (please doublecheck).

Author Response

The manuscript is well written and summarizes the current state of this topic detailed and nicely.

Figure 1: Here, a comment / hint could be included that purified donor SSCs are often propagated and (genetically) manipulated in an in vitro culture system before they are transplanted into a recipient testis.

- We have added more description in the figure legend for a comprehensive explanation of SSCT process as follows,

Testicular cells are isolated from the testis of donor male animals by collagenase digestion. Afterward, spermatogonia are enriched through multiple cycles of differential plating. The purity of spermatogonia can be further enhanced through discontinuous Percoll density gradient centrifugation. The enriched spermatogonia are adjusted to suitable concentration for transplantation. To ensure an effective SSCT, ablation of endogenous SSCs of recipient males is required. In this regard, busulfan treatment is generally effective for mouse SSC ablation, but remains inefficient in large animals. Other ablative method, such as irradiation of testis, also has limited effect on endogenous SSC removal to support an effective SSCT. Recently developed genetic modification technology can create genetically sterile animals with a complete SSC ablation but preserving intact testicular supporting cell structure, therefore offering an alternative approach to generate SSC-ablated recipient males. In mice, transplanted cells are usually injected through efferent tubules, whereas rete testis is the easy route for injection of spermatogonia in some large animals. Successful SSCT can reestablish spermatogenesis of recipient animals to generate donor-derived sperm, which can fertilize eggs to produce offspring harboring donor gene alleles.

Not all abreviations used in the manuscript are explained (please doublecheck).

- Thank you for your comments. We have check all abbreviations carefully throughout manuscript to ensure the full terms are defined when the abbreviations are firstly used.

Round 2

Reviewer 1 Report

The authors addessed my concerns.

Author Response

Thanks you for your positive comments. We have also invited some native speakers to help edit the manuscript. 

Reviewer 3 Report

While the professionality of the authors didn’t be criticized, they should explain the necessity of this review, especially the difference between the existing reviews listed (Reproduction. 2014 Feb 3;147(3):R65-74., Annu Rev Anim Biosci. 2019 Feb 15;7:385-401., J Anim Sci Biotechnol. 2019 Jun 12;10:46. Hum Reprod Update. 2020 Apr 15;26(3):368-391. Cell Tissue Res. 2020 May;380(2):393-414.) and the authors’.

The authors listed two of their works in the response: Zhang M, Biol Open. 2021; Zhang X, Livestock Sciences, 2021. However, neither of them could be found in PubMed. One possible explanation is that they reversely listed the first author of the papers. The Zhang X’s paper (The Biol Open. 2021 Jan 6;10(1)) is not related to large animals. The Zhang M’s paper (Livestock Science. 2021 Mar, 104448) is very similar to a previously published work (Theriogenology. 2017 Feb;89:365-373), which should be mentioned in the review.

Author Response

----Yes. I should elucidate the necessity to prepare this review. The current situation of this field is that almost no any significant progress are achieved and most previous publications cannot provide unequivocal evidences that SSCT is done in large animals. Among the technical problems of SSCT in large animals, I think the most significant limiting factor is how to effectively and maximally ablate endogenous germ cells of recipient male animals. This issue always interferes with the previous works from implantation of foreign SSCs to detection of transplanted SSCs. In this regard, traditional ablative methods such as busulfan and irradiation which remains in trial in large animals to date has to be abandoned or greatly optimized, as many evidences have proven their inefficiency in large animals. Therefore, the most feasible strategy is using genetically sterile recipients like the mice with W genotypes in 1980s when the SSCT in mice was firstly successful.

I think the most solid evidence for SSCT in large animals is JM Oatley’s work in PNAS, 2020, which uses the Nanos2 KO recipients. From this genetically sterile recipient, we can at least find that SSCT in large animals is practicable and can reestablish spermatogenesis. This point is our main idea and also shown in our review. Actually, we are working with the same idea which was not inspired by JM Oatley’s publication (using ETV5 KO to establish genetically sterile recipients for SSCT). This gene may not work well as SSCT effect seems limited, but we still work in many other genes using pig models.

We are explaining these as we believe SSCT in large animals has to use new strategy, thus some previous reviews could not give the most correct direction to this field. We also admit Oatley’s review (Annu Rev Anim Biosci. 2019 Feb 15;7:385-401) is a good reference source to our work.

----I am sorry to reversely list the authors of the two publications. The Biol Open paper reflects our idea in this field. We first use mouse models and then will try to translate to pigs. The Livestock Science paper is just published recently, later than submitting this review, thus was not included as a reference.

This manuscript is a resubmission of an earlier submission. The following is a list of the peer review reports and author responses from that submission.

Round 1

Reviewer 1 Report

In this review, Zhao and colleagues aim to summarize the current state of methodology for transplanting spermatogonial stem cells in large animals and the utility of this concept. While the review is comprehensive, the addition of insight for the field is incremental in light of several other publications on the same topic over the last 3-5 years. The authors have not really expanded on the topic other than to add discussion about recent studies of Ciccarelli et al., 2020, PNAS. 

There are several aspects that the authors should consider to improve the merit of the review:

The authors should reconsider some of the references used to support statements about SSCs. For example, on line 29 the referenced book chapter does not go into details about the role of SSCs in supporting high efficiency of spermatogenesis. Also, on line 33 none of the references are to primary research that show SSCs are the foundation for spermatogenesis; rather, all are review articles themselves which is acceptable as long as primary research publications are also provided. This is a problem in much of the introduction, as the authors simply reference to other review articles which makes the manuscript under consideration a review of reviews rather than a review of research.

Lines 38-40: the referenced publication does not provide any evidence that implanted SSCs can undergo the process of spermatogenesis. Primary research must be cited to support the statement.

Lines 47-49: to state that SSCT was performed in all of the species listed implies that the studies showed unequivocally that sperm with donor genetics were generated in the recipient testes. Unfortunately, very few of the referenced studies showed this to be true with direct experimental evidence. Thus, it is not entirely accurate to claim that SSCT was achieved in all of the species that have been listed.

Lines 61-85: nearly all of the description of spermatogenesis is for mice. Many of the aspects, such as As, Apr, and Aal may be specific to rodents and not relevant to large animals. The section should be re-written with a focus on what is known about spermatogenesis in large animals.

Line 111: GFRa1 is not a growth factor, rather it is a GPI anchored co-receptor and by itself does not stimulate SSC growth.

Lines 124-139: none of the referenced studies provided empirical evidence of in vitro maintenance or growth of large animal SSCs. The only way a cell can truly be identified as an SSC is if spermatogenesis can arise from it. Unfortunately, none of the referenced studies showed that any of the cultured cells could serve as a starting point for spermatogenesis. The authors' description of these studies should be reconsidered for accuracy.

Lines 140-206: in regards to livestock, neither use of chemotherapeutics nor irradiation nor heat shock are feasible or useful strategies to prepare recipients for SSCT. In reality, the only approach that could be of major utility is genetic strategies and in this regard, only knockout of NANOS2 has been shown to be a feasible approach in multiple livestock. Thus, the authors should reconsider the amount of attention paid to the other approaches when considering that they have little to no utility going forward when NANOS2 knockout has now been demonstrated.

Lines 232-245: all of the research conducted on mechanisms of SSC homing is with mouse models and the information may or may not have relevancy to large animals. The authors should temper how the findings with mice are consider in a review that is intended to focus on large animals.

Lines 265-273: discussion on use of PKH and CFDA dyes to label donor cells for detection after transplantation should be removed. These approaches are not scientifically sound when it come to clearly detecting spermatogenesis arising from transplanted donor SSCs.

Lines 276-278: the authors should reconsider how the utility of detecting GFP DNA in ejaculates of recipients is discussed as a means to determine success of SSCT. Sperm are not the only cells that can be present in an ejaculate, thus simply detecting GFP DNA does not unequivocally show that it was from sperm derived from transplanted SSCs. Also, in many of the studies referenced, proper controls to show that no residual lentivirus was present in the injected cell suspension were not included. Thus, ruling out that endogenous germ cells of the recipients were transduced with residual virus and therefore were the source of any possible sperm containing GFP is not possible. The authors should temper the discussion of this approach and outcomes of the studies that employed it.

The descriptions of transplantation results in Table 1 should be reconsidered for accuracy. In line with the point made above, the reference to any data derived from use of PKH and CFDA dyes should be avoided as this approach can generate misleading and inaccurate information. Thus, referenced studies 10, 15, 18, 19, 20, 129, 22, and 132 should be removed as the data are not definitive of SSC engraftment after transplantation of a mixed suspension of testis cells. Histological assessment of spermatogenesis in recipient testes that were not fully ablated of endogenous germline is not a valid approach to determining donor-derived spermatogenesis following presumed SSCT; one cannot unequivocally distinguish between donor and recipient using this approach. Thus, references 14 and 30 should be removed. Because sperm are not the only cells in an ejaculate, genotyping for microsatellite markers or transgene DNA is not definitive evidence of sperm derived from transplanted donor SSCs. If sperm are specifically isolated from the ejaculate or embryos are generated from the sperm and genotyped, the evidence can be compelling. Thus, only studies 11, 17, and 110 should be referenced as having provided valid evidence of donor sperm production following SSCT in large animals. Note that this reviewer cannot find the Kim et al., 2014 [reference 12] study in the peer reviewed literature.

The authors should reconsider including outcomes of SST in monkeys as it detracts from the central focus of the review being on utility of the approach in livestock to improve animal breeding. The utility of SSCT in monkeys is as a model for development of strategies that address human infertility; this topic could be an entire review by itself.

In section 7, "Attempts of SSCT in Large Animals", I recommend that the authors remove discussion of studies that solely used PKH or CDFA dyes or histological analysis or detection of microsatellite markers or transgene DNA as determinants of success. None of these approaches provide unequivocal evidence of donor sperm production from transplanted SSCs.

The writing needs to be improved throughout the manuscript as there are many missuses of the English language and confusing descriptions. For example, line 30, SSCs reside "on" not "in" the basement membrane.

Lastly, I caution the authors about duplicating the writing of previously published studies. Although not copied word for word, there are major portions of text in the submitted manuscript that are very similar in structure and meaning to other publications already in the peer-reviewed literature.

Reviewer 2 Report

Zhao et al. Spermatogonial stem cell transplantation in large animals

The authors reviewed the current state-of-the-art in spermatogonial stem cell transplantation with a focus on large mammals. In general, this is a good overview over the status of this field, and the chances and limitations are critically discussed. Spermatogonial stem cell transplantation may become a practically applicable technology for the concept of using surrogate sires in commercial cattle breeding, if the current bottlenecks can be solved.

Thus it would be important that the authors add a chapter on the immunology aspects of spermatogonial stem cell transplantation (SSCT). It is commonly accepted that the testes may represent an immune-privileged organ, however, a closer look-up would be helpful, since the translation of the SSCT from mice to livestock will require allogeneic transplantations. In the mouse model, allogeneic SSCT seems to result in drastically reduced fertility of the recipient! Are there any studies focusing on specific immune reactions, long-term stability, subchronic or chronic rejections?

For clarity and to inform the reader at a glance in Tab.1  additional rows of the most relevant outcome should be added: mature spermatozoa and concentration, fertilization capability by ICSI, IVF, artificial insemination or natural service, and number of obtained offspring.

In Tab. 1, goat, reference #110, it should be indicated that the assay shown in the paper do not un-equivocally prove that the spermatozoa were donor derived.